# Study on the Impact-Induced Energy Release Characteristics of Zr_68.5_Cu_12_Ni_12_Al_7.5_ Amorphous Alloy

**DOI:** 10.3390/ma14061447

**Published:** 2021-03-16

**Authors:** Jian Tu, Liang Qiao, Yu Shan, Chunliang Xin, Jiayun Liu

**Affiliations:** 1School of Mechatronical Engineering, Beijing Institute of Technology, Beijing 100086, China; tj2779@163.com; 2Department 2, Beijing Institute of Space Long March Vehicle, Beijing 100076, China; lyqiaoliang@hotmail.com (L.Q.); danbomingzhi2002@163.com (C.X.); jiayun@bit.edu.cn (J.L.)

**Keywords:** amorphous alloy, impact-induced reaction, shock compression, energetic structural materials

## Abstract

As a new kind of multifunctional energetic structural material (MESM), amorphous alloy will undergo a chemical reaction and release energy under impact load. In this paper, an analysis method for the impact-induced reaction parameters of solid materials was derived based on a three-term equation of state and Avrami–Erofeev equation. The relation between the degree of reaction, pressure, and temperature of Zr_68.5_Cu_12_Ni_12_Al_7.5_ amorphous alloy was obtained. The influence of participation of an oxidizing reaction on the material energy release efficiency was analyzed. The relation between the energy release efficiency and impact velocity was achieved by an experiment in which Zr_68.5_Cu_12_Ni_12_Al_7.5_ amorphous alloy fragments impact a steel plate. The variations of pressure and temperature during the impact process were obtained. In the end, a reaction kinetic model was modified, and the kinetic parameters for the impact-induced reaction of materials in an air environment were obtained.

## 1. Introduction

Multifunctional energetic structural materials (MESMs), also known as energetic metal materials or reaction metal materials, were firstly proposed by Montgomery and Hugh [1] in the name of reactive fragment. When the MESMs fragments strike the target at a certain velocity, besides the kinetic energy penetration ability of an inert fragment, a chemical reaction will occur under the impact pressure. It will release a lot of heat and generate severe explosion and combustion effects, which can improve the damage effect on the target, especially for the object behind the target.

As a result of its excellent properties and good application prospects, MESMs have been widely concerned and vigorously studied all over the world [1,2,3,4,5,6,7,8,9,10]. Wang et al. [6] have done a lot of theoretical and experimental research on the impact-induced reaction characteristics of fluoropolymer-based energetic reaction fragments. Their research contents are about Al/PTFE et al. energetic materials, involving the penetration behavior, fragmentation characteristics, and structure responses of projectiles and targets with different structures. They tested the quasi-static pressure in a closed container that was behind the target impacted by the fragment. Zhang et al. [7,8] and Xiong et al. [9,10] systemically studied the impact-induced energy release characteristics of Al/Ni-based MESMs. A thermochemical model based on a temperature-induced chemical reaction was deduced. They revised the thermal chemical reaction model of Al/Ni-based material according to the pressure results from fragment impact-induced reaction experiments.

The reactive fragment research mostly focuses on the field of polymers. This kind of material is made of metal powder and fluoropolymer. Generally, the powder is mixed and compressed to form the reactive material. Its advantage is that the material’s energy is high, while its strength is low. It can not be directly used as the fragment for a warhead. Therefore, the reactive material is usually filled in the metal shell to form a “metal capsule” to adapt to the high-pressure environment produced by detonation. However, the production efficiency is low due to the complex manufacture process.

Amorphous alloy is a new kind of MESM. As a result of its metallic properties, its strength is much higher than that of fluoropolymer. It has strong adaptability to the detonation environment and high penetration ability. The system of amorphous alloy particles is metastable in energy. Under high temperature or pressure, amorphous alloy will have a crystallization transition and release heat. At the same time, a chemical reaction will further release the energy. In this case, the fragment will impact the target with explosion/combustion reaction. Since amorphous alloy is formed by melting and casting, it also has the characteristics of a simple manufacturing process—in other words, a high production efficiency.

The primary disadvantage of amorphous alloy is that the main contribution of the energy released comes from an oxidation reaction, and the energy released by the chemical reaction is relatively low. Therefore, the energy release efficiency of amorphous alloy has always been a research hotspot in the field of damage. In this paper, Zr_68.5_Cu_12_Ni_12_Al_7.5_ amorphous alloy was studied. Based on the impact-induced temperature rise model and reaction kinetic model derived in this paper, the relation between the degree of reaction, impact pressure, and temperature of the material was studied. Experiments featuring fragments striking the steel target were conducted with a ballistic gun. The pressure generated in the quasi-closed container that was behind the target plate perforated by the fragment was measured. The energy release efficiency of Zr_68.5_Cu_12_Ni_12_Al_7.5_ amorphous alloy under impact environment was calculated. The change of temperature and pressure during the impact process was obtained. The reaction kinetic model was modified.

## 2. Theoretical Model for Impact-Induced Temperature Rise

The equation of state for solid materials is generally in the form of three terms:(1)$$P(V,T)=P_{c}(V)+P_{n}(V,T)+P_{e}(V,T)$$
where *P_c_* is the pressure at the temperature of 0 K, i.e., cold pressure; *P_n_* is the contribution of lattice thermal vibration to pressure; *P_e_* is the contribution of electron thermal motion to the pressure.

Based on Born–Meyer model, cold pressure can be expressed as
(2)$$P_{C}\left(\delta \right)=Q\delta^{2/3}\left\{\exp\left[q\left(1-\delta^{-1/3}\right)\right]-\delta^{2/3}\right\}$$
where *δ* = *ρ*/*ρ*_0*K*_ is the compression ratio; *Q* and *q* are cold energy parameters, which can be calculated by the analytical method shown in Ref [11]:(3)$$q=3\gamma_{0}+\sqrt{9\gamma_{0}{}^{2}-12\gamma_{0}+6}$$
(4)$$Q=\frac{3\rho_{0K}C_{0}{}^{2}}{q-2}$$
where *C*_0_ and *γ*_0_ are volume sound velocity at zero temperature and material constant, respectively; *ρ*_0*K*_ is the density under zero temperature and zero pressure, which can be calculated by Ref [12]
(5)$$\rho_{0K}=\rho_{0}\left[1+\frac{E_{D}\left(\Theta_{D}/T_{0}\right)}{C_{D}\left(\Theta_{D}/T_{0}\right)}\alpha_{0}\right]\approx \rho_{0}\left(1+540\alpha_{0}\right)$$
where *ρ*_0_, *T*_0_, and *α*_0_ represent the density, temperature, and linear expansion coefficient under normal condition (i.e., the temperature is 293 K, and the pressure is 1 atm), respectively; *E_D_*, *C_D_*, and Θ*_D_* represent the Debye energy, Debye specific heat, and Debye temperature, respectively.

According to Debye solid model, the contribution of lattice thermal vibration to pressure can be expressed as [11]
(6)$$P_{n}=\frac{\gamma }{V}\frac{3RT}{\mu }D\left(\frac{\Theta_{D}}{T}\right)$$
where *γ* is the Grüneisen coefficient; *V* is the specific volume; *R* is the universal gas constant; *μ* is the molar mass; and *D*(*x*) is the Debye function, when *T* is greatly larger than Θ*_D_*, *D*(Θ*D*/*T*) ≈ 1.

The Grüneisen coefficient *γ* can be described by Dugdale–MacDonald formula:(7)$$\gamma \left(\delta \right)=\frac{1}{6}\frac{q^{2}\delta^{-\frac{1}{3}}\cdot \exp\left[q\left(1-\delta^{-\frac{1}{3}}\right)\right]-6\delta }{q\cdot \exp\left[q\left(1-\delta^{-\frac{1}{3}}\right)\right]-2\delta }.$$

Using free electron model, the pressure produced by the thermal motion of the electron is
(8)$$P_{e}=\frac{1}{4}\beta_{0K}\rho_{0K}\delta^{1/2}T^{2}$$
(9)$$\beta_{0K}\approx 160.7{\left(\frac{602.2}{\mu \rho_{0K}{}^{2}}\right)}^{1/3}.$$

Let *P* = *P_H_*, where *P_H_* is the Hugoniot pressure. Substituting Equation (2), (6) and (8) into Equation (1), the equation of state becomes
(10)$$P_{H}=Q\delta^{2/3}\left\{\exp\left[q\left(1-\delta^{-1/3}\right)\right]-\delta^{2/3}\right\}+\frac{\gamma }{V}\frac{3RT}{\mu }D\left(\frac{\Theta_{D}}{T}\right)+\frac{1}{4}\beta_{0K}\rho_{0K}\delta^{1/2}T^{2}.$$

According to above formulas, the impact temperature *T* can be calculated with *P_H_*. *P_H_* can be gained with a Hugoniot energy equation and Grüneisen equation of state:(11)$$P_{H}\left(V\right)=\frac{\frac{V}{\gamma \left(\delta \right)}P_{C}\left(V\right)-E_{C}\left(V\right)}{\frac{V}{\gamma \left(\delta \right)}-\frac{1}{2}\left(V_{0}-V\right)}$$
where *V*_0_ is the initial specific volume, and *E_C_* is the cold energy, which can be expressed as follows:(12)$$E_{C}\left(\delta \right)=\frac{3Q}{\rho_{0K}}\left\{\frac{1}{q}\exp\left[q\left(1-\delta^{-1/3}\right)\right]-\delta^{1/3}-\frac{1}{q}+1\right\}.$$

## 3. Reaction Kinetic Model and Calculation of the Reaction Degree

Assuming that the factors that induce chemical reactions in materials are only caused by temperature [13], the Arrhenius model [14,15] can be used to describe the reaction kinetic behavior of the material. The chemical reaction equation can be expressed as
(13)$$\frac{dy_{d}}{dt}=Ae^{-E_{a}/RT}\;f(y_{d})$$
where *y_d_* is the degree of reaction; *t* is the reaction duration; *A* is the pre-exponential Arrhenius constant; *E_a_* is the activation energy; *R* is the molar gas constant; and *T* is the absolute temperature.

The temperature rising reaction with a high rate of solid material can be described as n-dimensional nuclear/growth controlled reaction model proposed by Avrami-Erofeev [15]:(14)$$f\left(y_{d}\right)=n\left(1-y_{d}\right){\left[-\ln\left(1-y_{d}\right)\right]}^{\left(n-1\right)/n}$$
where *n* is the order of the reaction, and it is related to the reaction mechanisms.

Ortega et al. [16] assume that the degree of reaction is a linear function of time:(15)$$\frac{dy_{d}}{dt}=Ct$$
where *C* is constant. Substituting Equations (14) and (15) into Equation (13), we can get:(16)$$\frac{y_{d}{}^{1/2}}{n(1-y_{d}){\left[-\ln(1-y_{d})\right]}^{\left(1-\frac{1}{n}\right)}}=\frac{A}{{\left(2C\right)}^{1/2}}e^{\left(-E_{a}/RT\right)}.$$

The first-order derivative of temperature with respect to the degree of reaction “*y_d_*” can be obtained from the above formula:(17)$$\frac{dT}{dy_{d}}=\frac{RT^{2}}{E_{a}}\left[\frac{1}{2y_{d}}-\frac{n\ln\left(1-y_{d}\right)+n-1}{n\left(1-y_{d}\right)\left[-\ln\left(1-y_{d}\right)\right]}\right].$$

According to the McQueen mixing rule [17], the expression for pressure *P_r_* after the reaction can be deduced [18]:(18)$$P_{r}=P+\frac{y_{d}Q_{R}}{\frac{V}{\gamma }-\frac{V_{0}-V}{2}}$$
where *P* is the pre-reaction pressure, and *Q_R_* is the chemical energy released by the reaction of reactant per unit mass.

The rise of energetic materials temperature after partial reaction can be regarded as the superposition of temperature rise effect of impact and the energy released by the chemical reaction, i.e.,
(19)$$T_{r}=T+y_{d}Q_{R}/C_{V}$$
where *T* is the temperature before the reaction, and *C_V_* is the specific heat of the material.

The impact-induced chemical reaction of amorphous alloy Zr_68.5_Cu_12_Ni_12_Al_7.5_ is carried out according to the following equations, where the standard enthalpy of formation (ΔH) for the reaction product can be obtained from Ref [19]:
Al + Ni ⟶ AlNi_3_   ΔH = 152.9 kJ/mol(20)
Zr + O_2_ ⟶ ZrO_2_ + O_2_   ΔH = 1078.3 kJ/mol(21)
2Al + 3/2O_2_ ⟶ Al_2_O_3_   ΔH = 1676 kJ/mol(22)
Cu + 1/2O_2_ ⟶ CuO   ΔH = 157.2 kJ/mol.(23)

The heat of intermetallic chemical reaction *Q_CR_*, which is 0.077 KJ/g, can be calculated with Equation (20). Using Equations (21)–(23), the heat of metal oxidation reaction *Q_OR_*, which is 9.938 KJ/g, can be obtained. During the impact process, because of the lack of oxygen inside the material, only the surface of the material is in contact with oxygen, and only a part of material can be oxidized. Assuming that the degree of participation in oxidation reaction is “*x*”, then the expression for impact-induced reaction heat is as follows:(24)$$Q_{R}=Q_{CR}+xQ_{OR}.$$

The activation energy *E_a_* of the material can be obtained with the DSC (Differential Scanning Calorimeter) method. The DSC equipment’s brand is Netzsch 204F1, and it is made in Germany. Figure 1 shows the DSC curve for Zr_68.5_Cu_12_Ni_12_Al_7.5_ at different heating rates.

According to the Kinssinger equation, the activation energy of the material can be calculated:(25)$$\ln\frac{\beta }{T^{2}}=-\frac{E_{a}}{RT}+C$$
where *β* is the heating rate; *T* is the characteristic temperature corresponding to different heating rates; *R* is the gas constant; and *C* is a constant.

By linear fitting ln(*β*/*T*^2^) and (−1/*T*), the slope of a straight line can be obtained (=3.216 × 10^4^), as shown in Figure 2. Then, *E_a_* can be obtained (=267.4 KJ/mol) under the temperature of 717 K.

Assuming that the degree of participation in the oxidation reaction is 0, 0.05, 0.2, and 0.5, the relation between the degree of reaction, pressure, and temperature of the Zr_68.5_Cu_12_Ni_12_Al_7.5_ amorphous alloy were obtained by us, as shown in Figure 3 and Figure 4. The calculation parameters for Zr_68.5_Cu_12_Ni_12_Al_7.5_ amorphous alloy are shown in Table 1.

As can be seen above, when the material is under the impact loading, the temperature will rise, which can induce the chemical reaction. The release of energy from the chemical reaction causes further increases in pressure and temperature. The activation energy *E_a_* determines the reaction threshold of the material. Participation in the oxidation reaction has a significant influence on the results. Increasing the supply of oxygen during the material reaction process can greatly improve the energy release efficiency of the material.

## 4. Energy Release Experiment and Calculation of the Parameters for Impact-Induced Reaction

### 4.1. Energy Release Experiment

To verify the accuracy of the reaction kinetic model derived above, a series of secondary impact-induced reaction experiments were conducted with a quasi-closed pressure test vessel. The pressure in the vessel that was behind the target impacted by the fragment was measured. This test is a common method to test the energy release characteristics of MESMs. The relationship between the impact velocity of the fragment and the released energy could be obtained.

The quasi-closed reaction vessel with a volume of 27 L was approximately hemispherical. The fragment incident end of the vessel was sealed by a layer of 1.5 mm thick front steel sheet. A 20 mm thick steel target plate was fixed in the rear end of the vessel. The piezoresistive sensor was installed on the inner wall of the vessel to test the pressure–time curve inside the vessel.

The fragments were driven by a 14.5 mm caliber ballistic gun. The principle and layout of the test is shown in Figure 5. The amorphous alloy fragments used in the experiments were formed by melting and casting.

The peak value of fragment impact-induced reaction pressure measured in the vessel was recorded as Δ*P_m_*. According to Ames theory [20], the pressure value in the vessel could be converted into the total energy Δ*Q* in the vessel:(26)$$\Delta P_{m}=\frac{\gamma_{a}-1}{V_{E}}\Delta Q$$
where *V_E_* is the volume of the closed container; and *γ_a_* is the specific heat ratio of the air in the container, which takes the constant value of 1.4.

There are two sources of pressure Δ*P_m_* in the closed vessel. One is the heat released by the chemical reaction during the impact process of amorphous alloy materials, and the other is the kinetic energy *E_k_* of the fragments launched into the vessel. The energy-releasing efficiency during the fragment impact process can be obtained as follows:(27)$$y_{e}=\frac{\Delta Q-E_{k}}{Q_{R}}$$
where *Q_R_* is the heat released by the complete reaction of fragments, and it is 10.015 KJ/g. The residual velocity vs. for fragments entering the vessel can be calculated by the classical Thor equation [21]:(28)$$\begin{array}{l} v_{s}=v-0.3048\times 10^{c_{1}}\times {\left(61,023.75\times H\times A\right)}^{c_{2}}\\ \times {\left(15,432.1\times m\right)}^{c_{3}}\times {\left(3.28084\times v\right)}^{c_{4}} \end{array}$$
where “*v*” is the shooting speed; “*H*” is the thickness of the target; “*A*” is the impact area of the fragment; “*m*” is the mass of the fragment; and *c*_1_–*c*_4_ is the constant related to the material of the target plate. In this paper, the target was No.45 steel plate. Hence, *c*_1_ = 6.399, *c*_2_ = 0.889, *c*_3_ = −0.945, and *c*_4_ = 0.019. It is assumed that there is no mass loss when fragments perforate the steel plate and enter the vessel. It is also assumed that part of the steel plates (*m_t_*) with the same impact cross-section enter the vessel. The kinetic energy of fragments in the vessel can be calculated:(29)$$E_{k}=\frac{1}{2}(m+m_{t})v_{s}{}^{2}.$$

The test fragment was a ball with the diameter of 9.5 mm. The peak pressure during the tests and the calculation results for the relevant parameters are shown in Table 2. The test pressure curves are shown in Figure 6.

### 4.2. Calculation of Impact-Induced Reaction Parameters

It is assumed that the impact process of a fragment on the target is one-dimensional. Based on the continuity condition, when fragments impact the target plate, the pressure *P* of the contact surface between the fragment and target is equal. The velocity of particles in the target plate can be obtained by the following equation [22]:(30)$$U_{P2}^{2}(\rho_{02}S_{2}-\rho_{01}S_{1})+U_{P2}(\rho_{02}C_{02}+\rho_{01}C_{01}+2\rho_{01}S_{1}v)-\rho_{01}(C_{01}v+S_{1}v^{2})=0$$
where *U_P_* is the velocity of the particle; *v* is the impact velocity; *C_0_* and *S* are Hugoniot parameters; and subscript 1 and subscript 2 represent the parameter values of the fragment material and target material, respectively. The parameter values are shown in Table 3.

After obtaining *U_P_*_2_ according to Equation (30), the particle velocity and shock wave velocity in the fragment can be further calculated:(31)$$U_{P1}=v-U_{P2}$$
(32)$$U_{S1}=C_{01}+S_{1}U_{P1}$$
where *U_S_* is the shock wave velocity. The impact pressure can be calculated according to the momentum conservation equation:(33)$$P=\rho_{01}U_{s1}U_{P1}.$$

According to the above equations, the relevant reaction parameters during the impact process of fragment were calculated, as shown in Table 4. The relation between energy release efficiency, impact velocity, and impact pressure are shown in Figure 7.

It can be seen that when the fragment impact velocity is lower than 530 m/s, the fragment does not react. With the increase of the fragment impact velocity, the impact pressure and temperature of the fragment increase, thus promoting the energy release efficiency during the fragment reaction process. When the impact velocity is higher than 1411 m/s, the reaction energy release efficiency tends to be stable, which is 0.142, as shown in Figure 7. Under this situation, the reaction degree of fragments has reached 1, and the energy release efficiency no longer increases with the increase of impact velocity.

## 5. Modification of the Parameter for Reaction Kinetic Model

The results of energy release tests show that a chemical reaction has taken place when the temperature is 665 k, which is lower than the activation energy temperature 717 K. It shows that temperature is not the only factor that affects the chemical reaction of the materials. The increase of pressure will also promote the chemical reaction of materials. By fitting the curve shown in Figure 7, the temperature threshold (i.e., *T_cr_*) of material reaction is 618 K, and the pressure threshold (i.e., *P_cr_*) of material reaction is 8.1 GPa.

As can be seen from Figure 7, the maximum energy release efficiency of fragments is 0.142 in the air environment, and the corresponding reaction degree of material is 1. According to Equation (24), the participation degree of oxidation reaction (i.e., *x*) is 0.135. Then, according to the relationship *y_d_* = *y_e_*/0.142, the energy release efficiency *y_e_* of each test in Table 2 was converted into the corresponding reaction degree *y_d_*, as shown in Table 5.

Substituting the temperature “*T*” shown in Table 4 and the reaction degree *y_d_* in Table 5 into Equation (16), the kinetic parameter of impact reaction can be obtained, i.e., *E_a_* = 80 KJ/mol, *n* = 1.1. Then, the modified *T_cr_*, *E_a_*, *n*, and *x* were substituted into the reaction kinetic model. As a result, the relation between reaction degree and impact pressure, and the relation between reaction degree and impact temperature were obtained, as shown in Figure 8 and Figure 9; where *P* and *T* are the impact pressure and impact temperature before material reaction, respectively; *P_r_* and *T_r_* are the impact pressure and impact temperature after material reaction, respectively. It can be seen that the calculated curves of the modified reaction kinetic model are in good agreement with the experiment results.

## 6. Discussion

In this paper, the fragment energy release tests were carried out in the air environment, and the modified kinetic parameters of impact reaction, *T_cr_*, *E_a_*, *n*, and *x* were only applicable to the air environment. The kinetic model of impact reactions in oxygen-rich or oxygen-poor environments will be investigated in the future. A series of experiments will be carried out to obtain the kinetic parameters of impact reactions under the corresponding environmental conditions. A comprehensive study on the energy release law of impact reactions of Zr_68.5_Cu_12_Ni_12_Al_7.5_ amorphous alloy will be conducted.

## 7. Conclusions

(1)The impact-induced temperature rising model and reaction kinetic model for solid materials were derived based on three-term equation of state and Avrami–Erofeev equation. The relationship between the degree of reaction, pressure, and temperature of Zr_68.5_Cu_12_Ni_12_Al_7.5_ amorphous alloy was obtained. The results show that the participation of oxidation reaction has a significant effect on the energy release efficiency of the materials.(2)The relationship between the energy release efficiency and impact velocity of the Zr_68.5_Cu_12_Ni_12_Al_7.5_ amorphous alloy fragment was obtained by an impact-induced energy release experiment. The results show that the chemical reaction starts when the impact velocity of fragments reaches about 600 m/s in the air environment. When the impact velocity of fragments exceeds 1411 m/s, the energy release efficiency tends to be stable, and its maximum value is 0.142.(3)It is concluded that the impact reaction of Zr_68.5_Cu_12_Ni_12_Al_7.5_ amorphous alloy is affected by both temperature and pressure through the impact energy release tests. The kinetic model for the impact reaction of materials was modified by the experimental data. The kinetic parameters of impact reactions in an air environment were obtained. The activation energy *E*_a_ = 80 KJ/mol; the order of reaction (i.e., *n*) is 1.1; the degree of participation in oxidation reaction (i.e., *x*) is 0.135; the temperature threshold *T_cr_* of material reaction is 618K; and the pressure threshold *P_cr_* of material reaction is 8.1GPa.

## Figures and Tables

**Figure 1 materials-14-01447-f001:**
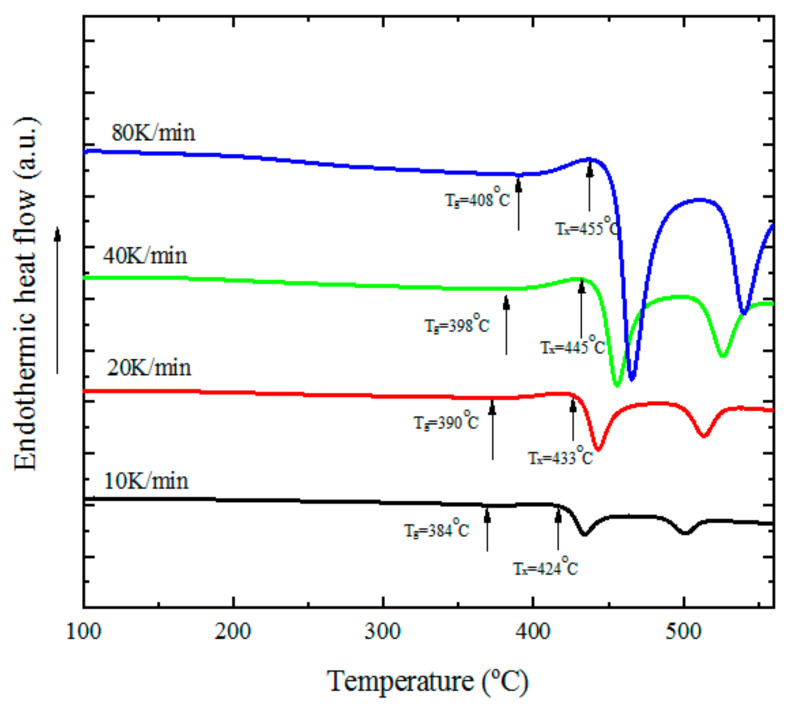
The DSC (Differential Scanning Calorimeter) curve for Zr_68.5_Cu_12_Ni_12_Al_7.5_ at different heating rates.

**Figure 2 materials-14-01447-f002:**
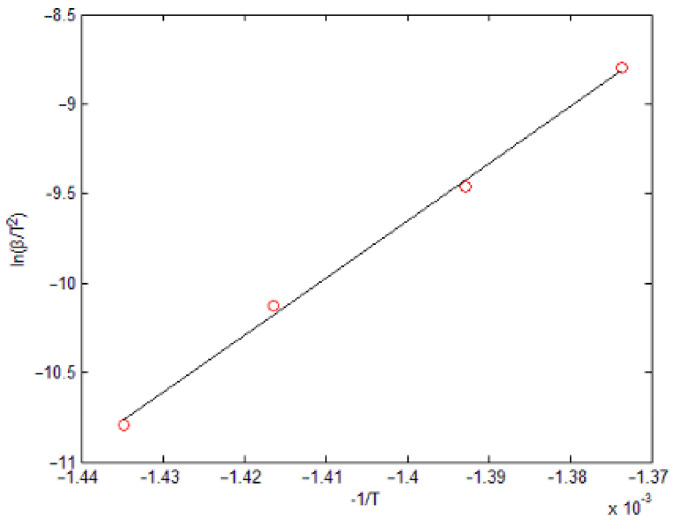
Fitting curve according to the Kinssinger equation.

**Figure 3 materials-14-01447-f003:**
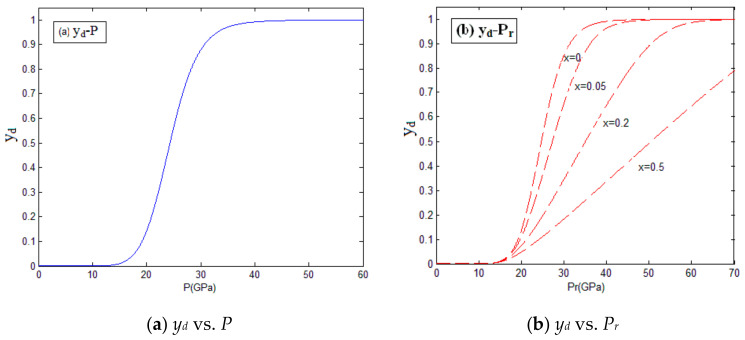
The relation between the degree of reaction and pressure. (**a**) The relation between the degree of reaction and pressure before material reaction; (**b**) The relation between the degree of reaction and pressure after material reaction.

**Figure 4 materials-14-01447-f004:**
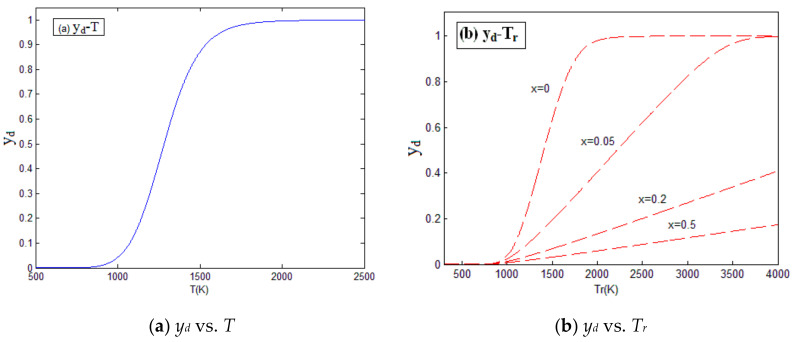
The relation between the degree of reaction and temperature. (**a**) The relation between the degree of reaction and temperature before material reaction; (**b**) The relation between the degree of reaction and temperature after material reaction.

**Figure 5 materials-14-01447-f005:**
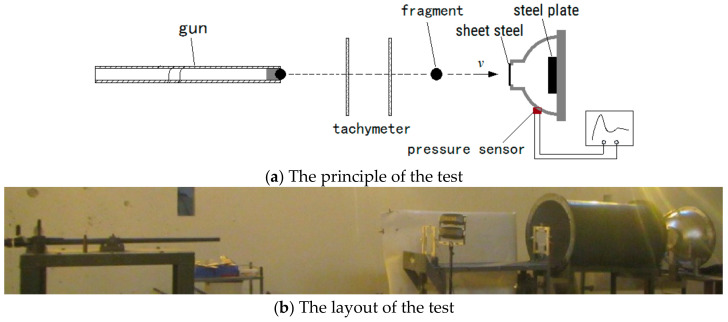
The principle and layout of the test site for fragment impact-induced reaction.

**Figure 6 materials-14-01447-f006:**
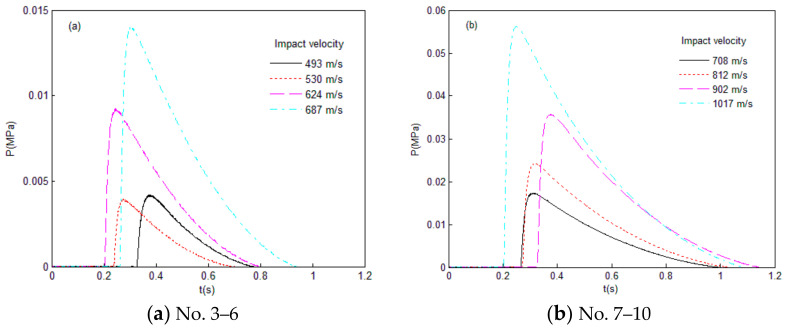
The test pressure–time curves for fragments with different impact velocity.

**Figure 7 materials-14-01447-f007:**
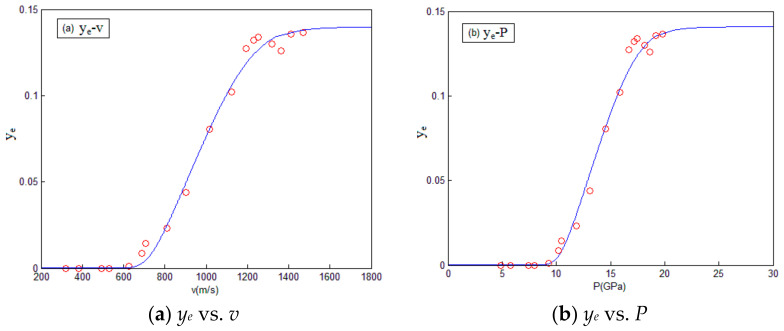
The relation between energy release efficiency, impact velocity, and impact pressure. (**a**) The relation between energy release efficiency and impact velocity; (**b**) The relation between energy release efficiency and impact pressure.

**Figure 8 materials-14-01447-f008:**
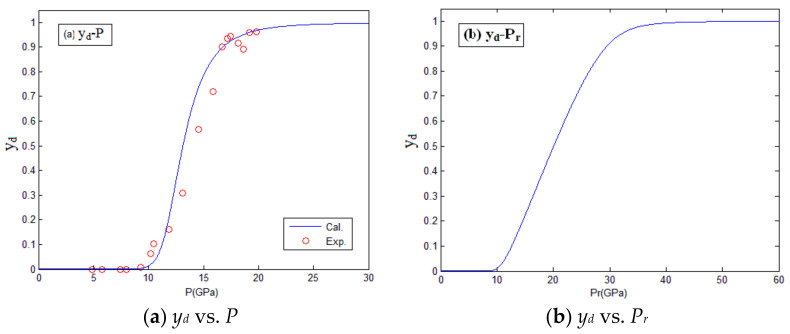
The relation between reaction degree and impact pressure. (**a**) The relation between the degree of reaction and pressure before material reaction; (**b**) The relation between the degree of reaction and pressure after material reaction.

**Figure 9 materials-14-01447-f009:**
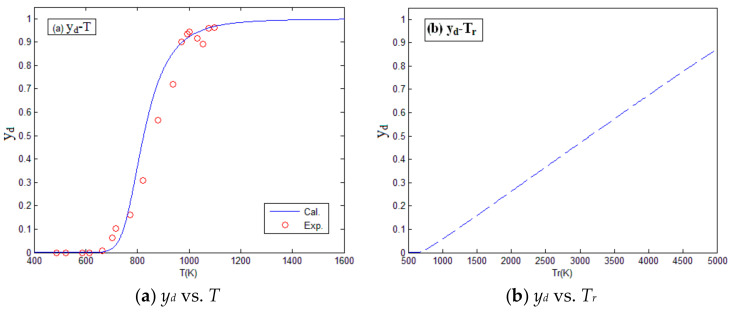
The relation between reaction degree and impact temperature. (**a**) The relation between the degree of reaction and temperature before material reaction; (**b**) The relation between the degree of reaction and temperature after material reaction.

**Table 1 materials-14-01447-t001:** The calculation parameters for Zr_68.5_Cu_12_Ni_12_Al_7.5_ amorphous alloy.

*ρ*_0_ (g/cm^3^)	*α*_0_ (10^−6^/K)	*C_V_* (J/g·K)	$\gamma_{0}$	*C*_0_ (km/s)	*Q* (GPa)	*q*	*n*
6.20	7.89	0.31	1.21	3.96	84.01	5.81	0.35

**Table 2 materials-14-01447-t002:** Fragment penetration test result.

No.	*m* (g)	*v* (m/s)	*v_s_* (m/s)	Δ*P_m_* (MPa)	Δ*Q* (kJ)	*E_k_* (kJ)	*y_e_*
1	2.79	391	320	0	0	0.15	0
2	2.77	453	382	0	0	0.22	0
3	2.79	564	493	0.004	0.27	0.36	0
4	2.77	602	530	0.004	0.27	0.42	0
5	2.81	696	624	0.009	0.61	0.58	0.001
6	2.8	759	687	0.014	0.95	0.70	0.009
7	2.78	780	708	0.017	1.15	0.74	0.015
8	2.76	884	812	0.024	1.62	0.98	0.023
9	2.8	974	902	0.036	2.43	1.21	0.044
10	2.74	1089	1017	0.056	3.78	1.53	0.080
11	2.8	1196	1124	0.070	4.73	1.87	0.102
12	2.76	1265	1192	0.084	5.67	2.11	0.128
13	2.8	1303	1230	0.088	5.94	2.25	0.132
14	2.77	1327	1254	0.090	6.08	2.33	0.134
15	2.78	1391	1318	0.092	6.21	2.58	0.130
16	2.79	1435	1362	0.093	6.28	2.75	0.126
17	2.77	1484	1411	0.100	6.75	2.95	0.136
18	2.73	1543	1470	0.104	7.02	3.21	0.136

**Table 3 materials-14-01447-t003:** The parameter data for calculation.

*ρ*_01_ (g/cm^3^)	*C*_01_ (km/s)	*S* _1_	*ρ*_02_ (g/cm^3^)	*C*_02_ (km/s)	*S* _2_
6.2	3.960	1.096	7.870	4.592	1.395

**Table 4 materials-14-01447-t004:** The relevant reaction parameters during the fragment impact process.

No.	*v_s_* (m/s)	*P* (GPa)	*U_P_*_1_ (m/s)	*T* (K)
1	320	4.9	172	487
2	382	5.8	203	523
3	493	7.4	256	586
4	530	8.0	274	614
5	624	9.3	317	665
6	687	10.2	344	704
7	708	10.5	353	716
8	812	11.9	397	773
9	902	13.1	433	823
10	1017	14.5	476	881
11	1124	15.9	515	939
12	1192	16.7	539	971
13	1230	17.2	552	992
14	1254	17.4	559	1000
15	1318	18.2	580	1033
16	1362	18.7	594	1054
17	1411	19.2	609	1075
18	1470	19.8	626	1099

**Table 5 materials-14-01447-t005:** The relationship between energy release efficiency *y_e_* and reaction degree *y_d_* obtained from the tests.

**No.**	**1**	**2**	**3**	**4**	**5**	**6**	**7**	**8**	**9**
*y_e_*	0	0	0	0	0.001	0.009	0.015	0.023	0.044
*y_d_*	0	0	0	0	0.007	0.063	0.106	0.162	0.310
**No.**	**10**	**11**	**12**	**13**	**14**	**15**	**16**	**17**	**18**
*y_e_*	0.080	0.102	0.128	0.132	0.134	0.130	0.126	0.136	0.136
*y_d_*	0.563	0.718	0.901	0.930	0.944	0.915	0.887	0.958	0.958

## Data Availability

The data used to support the findings of this study are available from the corresponding author upon request and within the article.

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
