# Peer review of "Study on the Impact-Induced Energy Release Characteristics of Zr68.5Cu12Ni12Al7.5 Amorphous Alloy"

_materials, 2021, doi:10.3390/ma14061447_

Round 1

Reviewer 1 Report

The authors of the paper: Study on the impact-induced energy release characteristics of ZrAlCuNi amorphous alloy present few interesting experimental results with theoretical considerations. Few minor suggestions can be take in consideration: 

L114: how was deltaH determined? experimentally or a reference is required. 

L192: what type of DSC equipment was used and the experimental conditions (temperature range, heating rate, correction used) 

L131: The figures 1 and 2 are made by the authors or a reference title is required 

L158-160: Figures 3 and 4 can be combined 

L162: instead of reference 17 the authors can use the source: 

Reference: R.G. Ames, Reaction efficiencies for impact-initiated energetic materials, 32nd International Pyrotechnics Seminar, June, 2005.

L204: Figure 4 - probably is Figure 6 - please check

Reviewer 2 Report

The paper “Study on the impact-induced energy release characteristics of ZrAlCuNi amorphous alloy” by Tu at al. deals with the characterization of a multifunctional energetic structural material based on ZrAlCuNi amorphous alloy. The authors perform energy released experiments, thus obtaining the relation between the degree of reaction, pressure and temperature. A modelling approach is used, and kinetic parameters for impact-induced reaction of materials in air are calculated.

The paper is well written, the conclusions are supported by data and the methodology used looks adeguate.

In the following I list some additional points that the authors should consider before publication:

lines 80-81: what do the authors mean by ‘normal density’ and ‘normal temperature’? What are the ‘normal conditions’ they are referring to? Different ‘normal conditions’ are used in different fields.

Eq. 11: please define the variable V_0

Eq. 15: what is A? A pre-exponential Arrhenius constant? Please clarify how to go from Eq. 13-14 to eq. 15.

Eq. 17: what is P? The pressure value before reaction? Similarly, is T in Eq. 18 the temperature value before reaction?

Table 1: what is a1?

How was the ZrAlCuNi amorphous alloy used for experiments obtained or produced? What was the composition of the fragments used in experiments? Details on this should be added to the manuscript.

Table 4: the y_e values (last column) are already shown in Table 2 (and are also slightly different, likely because of a different rounding). I would include them in one Table only, no need to repeat them twice.

Line 211: “[…] by the method of fitting curve” please refer to Figure 6 to better explain what you mean.

Table 5: please better clarify the relation between y_e and y_d, and explain the process to obtain y_d from y_e

Line 231: “Substitute […]” should be “Substituting […]”

Figure 8: please check the panels, they are not showing what they are supposed to show
